# TTRV: Test-Time Reinforcement Learning for Vision Language Models

## Abstract

Existing methods for extracting reward signals in Reinforcement Learning typically rely on labeled data and dedicated training splits, a setup that contrasts with how humans learn directly from their environment. In this work, we propose TTRV to enhance vision–language understanding by adapting the model on-the-fly at inference time, without the need for any labeled data. Concretely, we enhance the Group Relative Policy Optimization (GRPO) framework by designing rewards based on the frequency of the base model's output, while inferring on each test sample multiple times. Further, we also propose to control the diversity of model's output by simultaneously rewarding the model for obtaining low entropy of the output empirical distribution. Our approach delivers consistent gains across both object recognition and visual question answering (VQA), with improvements of up to $52.4\%$ and $29.8\%$, respectively, and average boosts of $24.6\%$ and $10.0\%$ across 16 datasets. Remarkably, on image recognition, TTRV applied to `Intern-VL-8B` surpasses GPT-4o by an average of $2.3\%$ over 8 benchmarks, while remaining highly competitive on VQA, demonstrating that test-time reinforcement learning can match or exceed the strongest proprietary models. Finally, we find many interesting properties of test-time RL for VLMs: for example, even in extremely data-constrained scenarios, where adaptation is performed on a single randomly chosen unlabeled test example, TTRV still yields non-trivial improvements of up to $5.5\%$ in recognition tasks.

> *"I never teach my pupils; I only attempt to provide the conditions in which they can learn."*
> — Albert Einstein

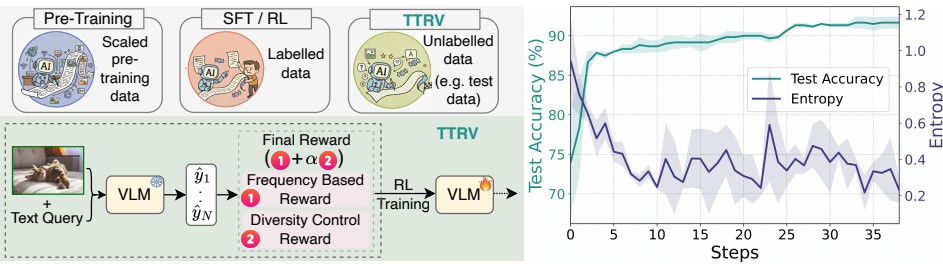

Figure 1: **Test-Time RL for VLMs.** (left) Unlike prior methods that require pre-training splits and post-training via Supervised Finetuning (SFT) or Reinforcement Learning (RL), our approach extracts reward signals directly at test time from unlabeled data. The reward combines (1) frequency-based signals and (2) diversity control, allowing the model to adapt online and improve downstream vision performance without any labeled data. (right) Test accuracy increases while entropy of the output logits decreases, showing that the model becomes more accurate and less uncertain as test-time RL progresses. The solid lines represent the mean, and shaded regions represent the variance of results obtained over 5 independent runs. The dataset is Resics45 (Cheng et al., 2017), task is object recognition, and the model is InternVL-3-2B (Zhu et al., 2025).

## 1 Introduction

Recent advances in vision–language models (VLMs) (Radford et al., 2021; Xu et al., 2023; Zhai et al., 2023; Liu et al., 2023b; Li et al., 2024; OpenAI, 2023) have enabled impressive progress on tasks such as

object recognition (Deng et al., 2009; Nilsback & Zisserman, 2008) and visual question answering (Fang et al., 2021; Yin et al., 2023). Yet, unlike humans, who continuously refine their reasoning by interacting with the world and adapting to ambiguous, unlabeled experiences, current VLMs remain largely static once trained. Adaptation typically requires large amounts of annotated data and costly fine-tuning, limiting their ability to cope with new domains or unseen tasks.

Reinforcement Learning (RL) has shown promise for improving reasoning in Large Language Models (LLMs) (Shao et al., 2024) and Vision–Language Models (VLMs) (Yu et al., 2025), and has emerged as an effective post-training method for enhancing task-specific performance. However, most existing approaches still rely on reward signals derived from human-labeled data and remain restricted to curated training splits, which are misaligned with real-world scenarios where train–test distinctions do not naturally exist. This dependence raises a fundamental question:

> *If RL is to embody true learning from experience, should it not arise directly from interaction with unlabeled data in the wild, rather than from curated benchmarks?*

In this work, we move toward this vision by proposing a **T**est-**T**ime **R**einforcement Learning framework for **V**ision Language Models (TTRV) that learns directly from unlabeled test data. Our TTRV extracts reward signals for Group Relative Policy Optimization (GRPO) (Shao et al., 2024) directly on the test data, as it is encountered. Specifically, our proposed reward formulation consists of two distinct parts, based on frequency and diversity control of the pre-trained model's output for each test sample. The intuition is to encourage the model to frequently produce similar outputs for each test sample and reward the predictions of the model which are more frequent and at the same time, control the diversity of model's output by rewarding lower entropy of the output empirical distribution. An overview of our work, along with one optimization trajectory, is provided in Figure 1. This approach transforms static pretrained VLMs into dynamic systems capable of self-improvement at inference time, bringing RL for multimodal models closer to the human-like paradigm of learning through raw experience.

We extensively evaluate our TTRV across 16 datasets spanning two tasks: image recognition and visual question answering (VQA). These datasets cover a diverse range of domains, including fine-grained recognition, math reasoning, and general VQA. Our results show that TTRV consistently improves performance, generalizes across model families, and is remarkably data-efficient. For instance, when post-training the InternVL3 (Chen et al., 2024b) model on only 20 randomly sampled test images, GRPO achieves gains of up to $52.4\%$ ($42.3\%$ on large-scale ImageNet (Deng et al., 2009)). Similarly, on VQA benchmarks, TTRV boosts performance by as much as $28.0\%$ on AI2D (Kembhavi et al., 2016). Remarkably, our TTRV outperforms one of the strongest proprietary models (GPT-4o) by $2.3\%$ on average (over 8 datasets) for image classification, while remaining highly competitive for VQA. Beyond these gains, our ablation studies uncover several interesting properties of GRPO for vision–language understanding. Notably, GRPO improves cross-dataset generalization: training on one dataset can yield strong gains on a completely unrelated dataset. Further, even in extremely data-scarce scenarios, GRPO remains effective, achieving up to $5.5\%$ improvement from rewards extracted on a single randomly chosen example. These findings suggest that GRPO does not simply adapt to a dataset's distribution but instead activates latent capabilities already learned during large-scale pretraining.

To conclude, we summarize the contributions of our work as follows:

- We introduce the first test-time reinforcement learning framework for vision–language models, which can be bootstrapped to any pre-trained VLM. Powered by carefully designed reward formulations, our method adapts models on the fly without requiring supervised data, thus realizing the true promise of RL.

- Through extensive experiments on 16 diverse benchmarks, we demonstrate that our proposed TTRV provides consistent and substantial improvements across tasks, model families, and domains.

- Our ablation studies further uncover novel properties of GRPO for VLMs, such as effectiveness in extremely low-data regimes and cross-dataset generalization, opening up new directions for future research in test-time adaptation with reward-driven learning.

## 2  RELATED WORK

Our work is closely related to vision-language models and works that study RL-based fine-tuning and test-time training (TTT) for VLMs.

**VLMs.** Recent progress in vision–language modeling has led to two major families of approaches. The first is dual-encoder models, where separate vision and text encoders are trained jointly, typically in a contrastive setting. These models excel at recognition-oriented tasks, with representative examples including CLIP (Radford et al., 2021), ALIGN (Jia et al., 2021), OpenCLIP (Schuhmann et al., 2022), SigLIP (Zhai et al., 2023), and MetaCLIP (Xu et al., 2023), as well as numerous extensions for downstream applications (Mirza et al., 2024; 2023c; Doveh et al., 2023b;a; Lin et al., 2023; Mirza et al., 2023a; Pathak et al., 2025). The second family, often referred to as large multimodal models (LMMs), couples a vision encoder with a large language model (LLM), enabling open-ended multimodal reasoning for tasks such as captioning, visual question answering (VQA), and document understanding. Pioneering approaches in this direction include BLIP-2 (Li et al., 2023b), InstructBLIP (Dai et al., 2023), MiniGPT (Zhu et al., 2024; Chen et al., 2024a), and the LLaVA series (Liu et al., 2023b; 2024; 2023a; Li et al., 2024). More recent models have pushed these capabilities even further: Qwen-2.5 VL (Bai et al., 2025) advances visual understanding by supporting precise object localization, dynamic resolution processing and strong agentic capabilities such as tool execution. InternVL3 (Zhu et al., 2025) improves perception and reasoning through native multimodal pretraining and domain-specific data such as 3D scenes, GUIs, and video. Phi-3.5 Vision (Abdin et al., 2024) offers a lightweight but strong alternative with long-context reasoning (128K tokens), robust vision inputs (images, charts, documents), and improved alignment via preference optimization. Several recent studies (Doveh et al., 2024; Gavrikov et al., 2024; Lin et al., 2024; Huang et al., 2024; Mirza et al., 2025) have further enhanced these models through improved training or adaptation strategies. In this work, we target the most recent open-source LMMs and focus on improving their test-time adaptability for vision-centric tasks such as object recognition, which remains a key weakness highlighted in prior studies (Zhang et al., 2024b; Mirza et al., 2025).

**RL-based fine-tuning for VLMs.** RL has become a central paradigm for aligning large language models with human preferences and task objectives, with approaches such as RLHF (Ouyang et al., 2022) and DPO (Rafailov et al., 2023) improving safety, coherence, and instruction-following in both LLMs and VLMs. More recently, rule-based methods like GRPO (Shao et al., 2024) have demonstrated the feasibility of scaling RL to enhance reasoning capabilities. Building on this foundation, RL-based fine-tuning (RFT) has been extended to multimodal models across a variety of vision-driven tasks. For example, VLM-R1 (Shen et al., 2025), VisualThinker-R1-Zero (Zhou et al., 2025), and Perception-R1 (Yu et al., 2025) adapt RFT for open-vocabulary object recognition, spatial reasoning, and visual perception, respectively; CLS-RL (Li et al., 2025) applies RFT to few-shot image classification; while R1-VL (Zhang et al., 2025) and related efforts further refine multimodal reasoning. These works demonstrate that RL can significantly enhance vision-centric capabilities of VLMs, but they still rely on curated training splits or labeled feedback. In contrast, our work investigates how reinforcement learning can be performed at *test time*, directly from unlabeled test data, thus bringing RL for VLMs closer to human-like learning from raw experience.

**Test-time training (TTT).** TTT methods adapt model parameters at inference without requiring labeled test data, typically by optimizing surrogate objectives such as entropy minimization or auxiliary self-supervised losses (Sun et al., 2020; Gandelsman et al., 2022; Sun et al., 2024; Mirza et al., 2023b; 2022). This idea is also similar to the recently popular Test-Time Scaling paradigm for LLMs (Snell et al., 2024). While first explored in unimodal settings, TTT has since been extended to multimodal models, with most efforts focusing on dual-encoder VLMs (*e.g.,* CLIP (Radford et al., 2021)). Representative approaches include TPT (Shu et al., 2022), which tunes text prompts via entropy minimization on augmented inputs, with extensions such as DiffTPT (Feng et al., 2023) and C-TPT (Yoon et al., 2024) improving augmentation quality and calibration; and RLCF (Zhao et al., 2023), which adapts the image encoder using feedback from larger models. Other works pursue lightweight black-box strategies that adapt embeddings without modifying internal parameters.

Most closely related to our work, TTRL (Zuo et al., 2025) introduces the idea of using reinforcement learning at test time for LLMs, where the majority voting across sampled outputs serves as a surrogate reward. While sharing the same high-level motivation, our TTRV differs greatly by extending the paradigm to multimodal models and combining frequency-based rewards with entropy regularization to balance consistency and diversity in predictions. The novel reward formulation helps us achieve better performance than the naïve majority voting (*c.f.,* Ablations Section 4.3). Further, in contrast to VLM-based approaches

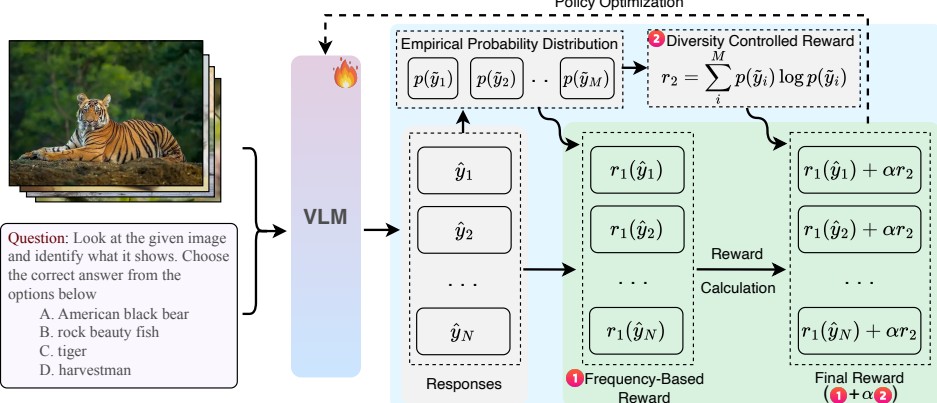

Figure 2: **Overview of TTRV.** For each prompt $x$, the VLM generates $N$ candidate responses $\{\hat{y}_1,...,\hat{y}_N\}$ from its policy $\pi_\theta(\cdot|x)$. These samples induce an empirical distribution over the unique outputs $\{\tilde{y}_1,...,\tilde{y}_M\}$, from which two reward signals are derived: (i) a *frequency-based reward*, where each response $y_j$ is rewarded in proportion to how often its output occurs among the $N$ responses (*i.e.,* its empirical probability in the distribution), and (ii) a *diversity control reward*, computed from the distribution to regulate diversity and encourage convergence. The final reward is the weighted combination of these terms, which is used to update the policy via GRPO.

that primarily target dual-encoder VLMs or prompt-level adaptation, our work focuses on decoder-based VLMs. To the best of our knowledge, our TTRV is the first framework to leverage GRPO for test-time RL of VLMs. Adapting models through RL at inference presents unique challenges, particularly in designing reward functions that operate in a fully unsupervised setting. We address this by rewarding predictions based on their frequency among the model's own outputs, while simultaneously regularizing diversity by rewarding model's certainty obtained by calculating the entropy of the empirical probability distribution. This formulation enables TTRV to achieve consistent improvements across diverse tasks and benchmarks.

## 3 TTRV: TEST TIME RL FOR VLMS

The goal of our proposed TTRV is to improve downstream vision tasks by extracting reward signals directly from unlabeled test data as it is encountered. To this end, we bootstrap an off-the-shelf VLM (*e.g.,* InternVL (Zhu et al., 2025)) with Group Relative Policy Optimization (GRPO) (Shao et al., 2024). A key contribution of our work lies in the design of fully unsupervised reward signals. At a high level, we introduce two complementary rewards: (i) a frequency-based reward that encourages consistent answers from the base VLM, and (ii) an entropy-based reward that regularizes the diversity of responses. An overview of our approach is provided in Figure 2 and a comprehensive python-like pseudo-code is provided in Appendix E, while the codebase is provided as a supplementary .zip file for review.

For ease of assimilation, the following subsections first provide a brief recap of Group Relative Policy Optimization (Section 3.1), then describe our proposed reward formulations in detail (Section 3.2), and finally present the resulting optimization objective for our TTRV (Section 3.3).

### 3.1 RECAP: GROUP RELATIVE POLICY OPTIMIZATION

Let $\mathcal{S}$ denote the space of natural language token sequences. A decoder-based vision-language model $\pi(\cdot|x)$, given an input prompt (image and text) $x \in \mathcal{S}$, produces a probability distribution over possible outputs $y \in \mathcal{S}$, where $y = (y_1, y_2,...,y_T)$ denotes a sequence of tokens. The probability of generating a sequence $y$ is $\pi(y|x) = \prod_{t=1}^{T} \pi(y_t \mid y_{<t}, x)$.

Post-training with reinforcement learning aims to maximize a scalar reward function $r : \mathcal{S} \times \mathcal{S} \to \mathbb{R}$ while constraining deviation from a reference policy $\pi_{\text{ref}}$. This leads to the KL-regularized optimization problem:

$$\max_\pi \mathbb{E}_{x \sim D, y \sim \pi(\cdot|x)} \Big[ r(x,y) \Big] - \beta D_{\text{KL}}\big(\pi(\cdot|x) \| \pi_{\text{ref}}(\cdot|x)\big), \tag{1}$$

where $D$ is the dataset of prompts, and $\beta > 0$ controls the KL regularization strength.

Group Relative Policy Optimization (GRPO) (Shao et al., 2024) provides a stable approach to optimize this objective. Given $n$ sampled responses $\{y_i\}_{i=1}^n$ for a prompt $x$, the advantage of each response is defined as

$$A_i = \frac{r(x,y_i) - \text{mean}_j(r(x,y_j))}{\text{std}_j(r(x,y_j))}, \tag{2}$$

which measures relative performance within the sampled group. The policy update is performed via clipped importance-weighted objectives to ensure stability, while KL regularization keeps the fine-tuned model close to the reference distribution.

## 3.2 TEST-TIME RL WITH DISTRIBUTIONAL REWARDS

Our TTRV extends the vanilla GRPO framework, which is usually applied by extracting rewards by using labeled data, by introducing an inference-time RL framework. We propose to extract self-supervised reinforcement signals directly from the empirical distribution of model outputs at inference time. Unlike settings that rely on external supervision, our framework generates self-consistent rewards that exploit the variability of rollouts to guide convergence during inference. In particular, we extract two rewards from the unlabeled data.

**Frequency-Based Reward.** Given a copy of the model at a particular time step during test-time learning, our goal is to infer on the test sample multiple times and reward predictions based on their frequency. The underlying intuition is that responses produced more consistently by the model are more likely to be correct. Formally, for each test sample $x$ (consisting of an image and text prompt), we sample $N$ candidate responses $\{\hat{y}_1, \hat{y}_2, ..., \hat{y}_N\}$ from the current policy $\pi_\theta(\cdot|x)$. Let $\mathcal{U} = \{\tilde{y}_1, \tilde{y}_2, ..., \tilde{y}_M\}$ denote the set of unique outputs. We estimate the empirical probability of $\tilde{y}_m$ as

$$p(\tilde{y}_m) = \frac{1}{N}\sum_{j=1}^N \mathbf{1}\{\hat{y}_j = \tilde{y}_m\}, \tag{3}$$

where $\mathbf{1}$ is an indicator function. The reward for an individual sample $\hat{y}_j$ is then defined as

$$r_1(\hat{y}_j) = \sum_{m=1}^M p(\tilde{y}_m) \cdot \mathbf{1}\{\hat{y}_j = \tilde{y}_m\}, \tag{4}$$

which assigns higher values to responses that occur frequently, while still allocating nonzero reward to less common but potentially meaningful alternatives. This graded structure captures the implicit consensus among repeated rollouts without discarding minority reasoning paths.

Importantly, this differs from the standard best-of-$N$ sampling scheme employed by (Zuo et al., 2025), which selects only the most frequent response and discards all others. Such a hard decision can be problematic when the model is uncertain or when the most frequent prediction is incorrect, since it provides a misleadingly strong but potentially wrong reward signal. In contrast, our reward formulation produces a *soft, probabilistic supervision signal* that reflects the full distribution over responses. This perspective is naturally connected to Bayesian reasoning: rather than collapsing to a single point estimate, our method retains uncertainty over hypotheses and uses it to shape learning. We further validate this design choice through ablations against naïve best-of-$N$ sampling, with results reported in Section 4.3.

**Diversity Control Reward.** Complementing the frequency-based reward $r_1$, which provides soft, frequency-proportional credit to repeated model responses, we introduce an entropy-based regularizer to control convergence. For a given test sample we compute the Shannon entropy (Shannon, 1948) of the empirical response distribution:

$$H(P) = -\sum_{m=1}^M p(\tilde{y}_m)\log p(\tilde{y}_m), \tag{5}$$

and define the auxiliary reward

$$r_2 = -H(P). \tag{6}$$

which penalizes excessive dispersion in the output distribution. This mechanism ensures that while the model explores diverse reasoning modes initially (as encouraged by the frequency-based reward), it gradually consolidates probability mass toward stable, high-probability answers rather than spreading attention too thin across redundant responses.

| | ImageNet | ImageNet-V2 | ImageNet-R | ImageNet-S | ImageNet-A | Food101 | DTD | Resisc45 | Mean |
|---|---|---|---|---|---|---|---|---|---|
| GPT-4o | 98.30 | 95.10 | 91.70 | 91.20 | 90.60 | 95.60 | 92.30 | 92.13 | 93.37 |
| CLIP | 68.33 | 61.86 | 76.90 | 48.27 | 49.91 | 87.04 | 44.73 | 58.22 | 61.91 |
| MetaCLIP | 70.78 | 62.64 | 80.99 | 57.91 | 46.75 | 85.48 | 55.80 | 66.19 | 65.82 |
| EVACLIP | 74.72 | 67.03 | 81.95 | 57.73 | 53.91 | 87.65 | 52.71 | 60.37 | 67.01 |
| SigLIP | 76.05 | 68.97 | 90.33 | 67.91 | 45.33 | 89.84 | 64.79 | 64.54 | 70.97 |
| LLaMA-3.2-11b | 72.68 | 39.94 | 72.26 | 69.39 | 83.07 | 93.99 | 87.54 | 82.51 | 75.17 |
| LLaVA-1.5-7b | 97.74 | 95.85 | 96.46 | 94.58 | 94.39 | 95.35 | 76.68 | 92.29 | 92.92 |
| Phi-3.5-vision | 97.94 | 95.66 | 96.05 | 94.93 | 85.78 | 96.10 | 88.26 | 85.89 | 92.58 |
| InternVL3-2B | 56.00 | 67.43 | 66.01 | 62.19 | 67.92 | 67.19 | 37.24 | 72.28 | 62.03 |
| w/ **TTRV** | 98.31 | **98.25** | **96.89** | 94.74 | 96.31 | 95.60 | **89.73** | 90.06 | 94.99 |
| Δ | +42.31 | +30.82 | +30.88 | +32.55 | +28.39 | +28.41 | +52.49 | +17.78 | +32.95 |
| InternVL2.5-4B | 93.26 | 83.07 | 79.53 | 65.51 | 90.67 | 80.92 | 47.33 | 23.44 | 70.47 |
| w/ **TTRV** | 97.11 | 95.66 | 88.21 | 92.01 | 96.00 | 94.49 | 81.98 | 13.30 | 82.34 |
| Δ | +3.85 | +12.59 | +8.68 | +26.50 | +5.33 | +13.57 | +34.65 | -10.14 | +11.88 |
| InternVL3-8B | 79.47 | 62.58 | 59.32 | 54.48 | 57.03 | 78.32 | 59.11 | 83.62 | 66.74 |
| w/ **TTRV** | **99.31** | 97.24 | 96.88 | **95.03** | **96.86** | **97.20** | 89.37 | **93.82** | 95.71 |
| Δ | +19.84 | +34.66 | +37.56 | +40.55 | +39.83 | +18.88 | +30.26 | +10.20 | +28.97 |

Table 1: **Image Classification.** Top-1 Accuracy (%) obtained by evaluating multiple different backbones. The results in gray are obtained using the specialized dual-encoder VLMs and the proprietary GPT-4o. For decoder-based VLMs we also evaluate multiple families and model sizes. Our TTRV is applied to different model sizes from the InternVL (Zhu et al., 2025) family of models. The best results obtained for a dataset are highlighted in **bold**, while the second best are underlined.

**Combined Reward.** The overall reward assigned to a response $\hat{y}_j$ is the combination of probability and entropy terms:

$$R(\hat{y}_j) = r_1(\hat{y}_j) + \alpha r_2, \tag{7}$$

where $\alpha$ is a tunable hyperparameter controlling the trade-off between convergence and diversity. By combining probability-based self-rewarding with entropy regularization, the model adaptively aligns its outputs during inference, striking a balance between exploring diverse reasoning paths and converging to coherent predictions.

## 3.3 OPTIMIZATION OBJECTIVE

The reinforcement learning objective is to maximize the expected reward under the policy:

$$\max_{\theta} \mathbb{E}_{y \sim \pi_\theta(\cdot|x)}[R(y)]. \tag{8}$$

For decoder-based VLMs, optimization is performed through the standard autoregressive language modeling objective, with the reward providing a soft, sample-level weighting of predicted tokens. The parameters are updated via gradient ascent:

$$\theta \leftarrow \theta + \eta \nabla_\theta \mathbb{E}_{y \sim \pi_\theta(\cdot|x)}[R(y)], \tag{9}$$

where $\eta$ denotes the learning rate.

We note that GRPO (Shao et al., 2024) modifies this process by replacing the raw reward with a relative advantage term (as defined in equation 2). This shifts optimization from absolute rewards toward relative comparisons, making it more stable and better aligned with group-level objectives.

## 4 RESULTS

In this section, we first list the implementation details, which include an introduction to the datasets used for evaluating our TTRV, then provide an overview of the different baselines we compare to, and finally conclude with a discussion of the main results and ablations. The details about implementation and evaluation protocols are delegated to the Appendix Section A.

## 4.1 EVALUATION SETTINGS

**Image Recognition Datasets.** We evaluate on eight diverse object recognition benchmarks. These include the two original ImageNet test sets: ImageNet (Deng et al., 2009) and ImageNet-V2 (Recht et al., 2019), along with three out-of-distribution variants: ImageNet-Rendition (R) (Hendrycks et al., 2021a), ImageNet-Sketch (S) (Wang et al., 2019), and ImageNet-Adversarial (A) (Hendrycks et al., 2021b). In

|  | **Mathverse** | **Mathvista** | **SEED** | **MME** | **RealWorldQA** | **Capture** | **CRPE** | **AI2D** | **Mean** |
|---|---|---|---|---|---|---|---|---|---|
| GPT-4o | 54.40 | 63.80 | 69.80 | 89.75 | 75.40 | 85.25 | 76.60 | 84.60 | 71.97 |
| LLaMA-3.2-11b | 19.36 | 35.40 | 62.56 | 51.59 | 41.18 | 61.48 | 44.43 | 59.54 | 53.34 |
| LLaVA-1.5-7b | 26.02 | 34.29 | 61.50 | 49.05 | 59.87 | 71.75 | 64.84 | 47.49 | 46.02 |
| Phi-3.5-vision | 36.02 | 51.22 | 69.54 | 77.22 | 57.25 | 72.99 | 68.33 | 75.55 | 48.58 |
| InternVL3-2B | 44.10 | 58.26 | 24.99 | 17.04 | 63.47 | 60.27 | 71.92 | 39.68 | 47.47 |
| w/ **TTRV** | 48.51 | 66.11 | 48.85 | 11.06 | 64.29 | 78.64 | 72.00 | 67.75 | 57.15 |
| Δ | +4.41 | +7.85 | +23.86 | -5.98 | +0.82 | +18.37 | +0.08 | +28.07 | +9.69 |
| InternVL2.5-4B | 51.69 | 65.49 | 57.37 | 85.27 | 65.25 | 80.03 | 74.33 | 51.55 | 66.37 |
| w/ **TTRV** | **53.02** | **66.94** | 61.14 | **85.79** | 66.00 | **85.99** | 75.22 | 61.09 | 69.40 |
| Δ | +1.33 | +1.45 | +3.77 | +0.52 | +0.75 | +5.96 | +0.89 | +9.54 | +3.03 |
| InternVL3-8B | 34.56 | 38.84 | 32.12 | 49.02 | 19.01 | 59.50 | 55.81 | 30.95 | 38.05 |
| w/ **TTRV** | 42.15 | 50.41 | 59.16 | 78.77 | 26.57 | 80.68 | 68.26 | 53.92 | 55.56 |
| Δ | +7.59 | +11.57 | +27.04 | +29.75 | +7.56 | +21.18 | +12.45 | +22.97 | +17.50 |

Table 2: **Visual Question Answering.** Results obtained by evaluating multiple different backbones. For decoder-based VLMs, we evaluate multiple families and model sizes. Our TTRV is applied to different model sizes from the InternVL (Zhu et al., 2025) family of models.

addition, we consider two fine-grained recognition datasets: Food101 (Bossard et al., 2014) and the Describable Textures Dataset (DTD) (Cimpoi et al., 2014), as well as a remote sensing dataset based on satellite imagery: Resisc45 (Cheng et al., 2017).

**VQA Datasets.** We further evaluate our TTRV on eight visual question answering (VQA) datasets covering a broad range of reasoning skills. These include two math reasoning benchmarks: MathVerse (Zhang et al., 2024a) and MathVista (Lu et al., 2024); three datasets focusing on everyday scenarios and objects: SEED (Li et al., 2023a), MME (Yin et al., 2023), and RealWorldQA (AI, 2024); two compositional reasoning datasets: Capture (Pothiraj et al., 2025) and Circular-based Relation Probing Evaluation (CRPE) (Wang et al., 2024); and one dataset targeting chart-based questions: AI2D (Kembhavi et al., 2016).

These 16 datasets were deliberately selected to span a broad spectrum of domains and tasks, including natural images, fine-grained categories, remote sensing, mathematical reasoning, everyday commonsense, compositionality, and chart understanding. This diversity ensures that our findings are not confined to a single domain but instead provide a representative view of model capabilities across varied and challenging settings. We further expect that the insights derived here will generalize to other benchmarks, which can be incorporated in future evaluations.

**Baselines:** For comparison, we evaluate the following dual-encoder VLMs: CLIP (Radford et al., 2021), MetaCLIP (Xu et al., 2023), EVACLIP (Sun et al., 2023), and SigLIP (Zhai et al., 2023). As representative methods for decoder-based VLMs, we choose: LLaMA (Touvron et al., 2023), LLaVA (Liu et al., 2023b), Phi-3.5-vision (Abdin et al., 2024), and also provide results for the proprietary GPT-4o (OpenAI, 2023). For the main experiments, we apply our TTRV to different model sizes of the InternVL (Chen et al., 2024b) family. However, we want to point out that TTRV can be applied to any open-source VLM. We provide results with QwenVL (Bai et al., 2025) in the ablations Section 4.3.

## 4.2 RESULTS

**Image Classification.** In Table 1 we present the top-1 accuracy across eight diverse image recognition benchmarks. We observe that TTRV consistently enhances performance for all evaluated InternVL backbones, with particularly strong gains on challenging distribution shifts such as ImageNet-R and ImageNet-S. For example, when applied to InternVL3-2B, TTRV improves accuracy by up to 89.7% on DTD and 90.0% on Resisc45, while yielding an average gain of around 32.9% across datasets. Similar trends appear for larger models: InternVL2.5-4B and InternVL3-8B see systematic boosts on both fine-grained recognition (Food101) and large-scale benchmarks (ImageNet and variants). Notably, TTRV lifts InternVL3-8B to over 99% accuracy on ImageNet, even outperforming proprietary systems (*e.g.,* GPT-4o) and establishing new state-of-the-art results for open-source VLMs. We emphasize that these improvements are achieved by randomly sampling only 20 test instances per dataset, suggesting that TTRV may not be adapting strongly to the test data distribution itself, but rather recovering and amplifying visual recognition capabilities that were present in pre-training and which might be attenuated during instruction tuning. We also observe some particular cases when the model performance decreases (*e.g.,* on the Resics45 dataset with InternVL-2.5-4B model). This can be attributed to the low performance of the base model,

|                              | Mathvista | SEED  | AI2D  |
|------------------------------|-----------|-------|-------|
| InternVL2.5-4B (base)        | 65.49     | 57.37 | 51.55 |
| w/ Maj Voting                | 65.08     | 58.37 | 47.52 |
| Δ vs. base                   | -0.41     | +1.00 | -4.03 |
| TTRV (w/o Freq. reward)      | 66.81     | 58.87 | 52.66 |
| Δ vs. base                   | +1.32     | +1.50 | +1.11 |
| TTRV (w/o Diversity reward)  | 65.08     | 59.27 | 53.06 |
| Δ vs. base                   | -0.41     | +1.90 | +1.51 |
| TTRV (Freq. + Diversity)     | 66.94     | 61.14 | 61.09 |
| Δ vs. base                   | +1.45     | +3.77 | +9.54 |

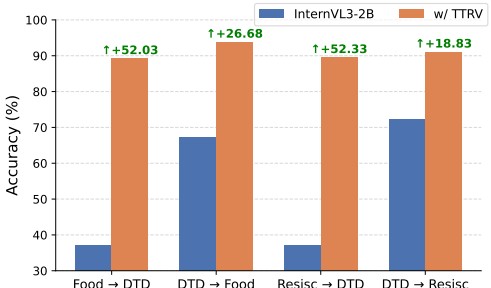

Table 3: **Ablating Reward Designs.** We compare the design choices of our TTRV with the reward design proposed by Zuo et al. (2025), based on the pseudo-labels obtained from a majority voting scheme. Further, we also ablate the individual effect of our frequency- and diversity-based rewards.

Figure 3: **Cross-dataset Generalization.** Top-1 accuracy (%) achieved by employing TTRV on a base dataset using InternVL3-2B and evaluating on a target dataset from a completely different domain. The results highlight that TTRV enhances core abilities of the model.

which might result in extremely low-quality rollouts from the model or general instability of GRPO (Wang et al., 2025). Overall, these findings demonstrate that frequency- and entropy-based test-time rewards allow models to consolidate predictions more effectively, leading to robust improvements in visual recognition.

**Visual Question Answering.** In Table 2 we report results across eight multimodal reasoning tasks, including mathematical problem solving (MathVerse, MathVista), scientific diagram understanding (AI2D), and general-domain evaluation (RealWorldQA). We find that TTRV provides consistent gains across all datasets and model scales. For instance, on InternVL2.5-4B, accuracy improves by 4.4% on MathVista and 9.5% on AI2D, while the larger InternVL3-8B benefits from 12.4% and 7.5% improvements on CRPE and RealWorldQA, respectively. We also observe that TTRV outperforms other open-source VLMs while also remaining highly competitive with GPT-4o (only lagging behind by $\sim 2\%$ on average), and outperforming the strong proprietary model on some benchmarks, like the challenging Mathvista. Furthermore, these gains are also obtained using only 20 sampled instances per dataset, suggesting that the improvements may not stem from distribution-level adaptation but rather from leveraging and re-aligning latent reasoning skills that were learned during pre-training but weakened after instruction tuning. This highlights that TTRV is especially effective at recovering such capabilities under test-time optimization, enabling more robust reasoning across diverse VQA tasks.

## 4.3 ABLATIONS

In this section, we present extensive ablations of our method and our design choices. We begin by comparing our reward formulation against the naïve best-of-N (majority voting) sampling strategy for RL. Next, we evaluate the robustness of TTRV by training on one dataset and testing on a completely different distribution. We then investigate alternative sampling techniques and reward designs, followed by experiments in extremely data-scarce settings where TTRV is applied to a single randomly chosen test sample. Finally, we report results obtained by applying TTRV to the Qwen-VL model, highlighting generalization of TTRV beyond the Intern-VL models. Due to space constraints, we delegate additional ablations (*e.g.,* on latency) to the Appendix.

**Reward Designs.** In Table 3 we ablate different reward designs. Specifically, we compare the reward design proposed in our work with the majority voting reward used by (Zuo et al., 2025), and also ablate the effect of the two different rewards used in our work (*c.f.,* Section 3). We find that the combination of the two rewards: frequency-based and diversity-control, outperforms all other design choices.

**Cross-Data Generalization.** In the main results (*c.f.,* Tables 1 & 2), we evaluate TTRV on test samples drawn from the same dataset (as used for test-time RL). In contrast, Figure 3 reports results when TTRV is applied using one dataset and evaluated on a completely different distribution (*e.g.,* Food101 for TTRV and DTD for testing). We observe that TTRV exhibits strong cross-data generalization, indicating that its performance gains stem not from distribution-specific adaptation but from enhancing the model's underlying task ability (*e.g.,* image classification).

**Effect of Data Sampling.** We find that TTRV does not require sampling data from all classes in the downstream dataset to achieve strong performance gains. In Table 4, we compare *biased sampling*, where

|  | Imagenet-A | Imagenet-R |
|---|---|---|
| InternVL3-4B (base) | 90.67 | 79.53 |
| w/ biased sampling | 95.09 | 88.51 |
| $\Delta$ vs. base | +4.42 | +8.98 |
| w/ random sampling | 96.00 | 88.21 |
| $\Delta$ vs. base | +5.33 | +8.68 |

|  | Seed | Imagenet-R |
|---|---|---|
| InternVL2.5-4B (base) | 57.37 | 79.53 |
| w/ Random Rewards | 52.41 | 78.00 |
| $\Delta$ vs. base | -4.96 | -1.53 |
| TTRV (Freq. + Diversity) | 61.14 | 88.21 |
| $\Delta$ vs. base | +3.77 | +8.68 |

Table 4: **Biased vs. Random Sampling.** Top-1 accuracy (%) obtained by sampling the test data differently. For biased sampling, we choose a fraction of the data from only a subset of classes (*e.g.,* 4 out of 200 for ImageNet-R). Random sampling results are obtained by sampling the data randomly from all classes.

Table 5: **Random Reward vs. TTRV.** We compare the results obtained by using random rewards (following Shao et al. (2025)) and our sophisticated reward design. The results highlight that the gains obtained with *spurious rewards* do not transfer to Intern-VL family of models, which were shown for Qwen-based models.

|  | Mathvista | SEED | Imagenet-A | Imagent-R |
|---|---|---|---|---|
| InternVL2.5-4B | 65.49 | 57.37 | 90.67 | 79.53 |
| w/ TTRV | 66.11 | 58.87 | 95.28 | 85.00 |
| $\Delta$ | +0.62 | +1.50 | +4.61 | +5.47 |

|  | Mathverse | Mathvista | Capture | Resisc45 |
|---|---|---|---|---|
| Qwen2.5-VL-3B | 45.33 | 67.35 | 71.33 | 90.08 |
| w/ TTRV | 48.71 | 71.48 | 75.25 | 92.71 |
| $\Delta$ | +3.38 | +4.13 | +3.92 | +2.63 |

Table 6: **Single Example TTRV.** We report results for VQA and image classification after applying TTRV on a single randomly sampled test example. The evaluation is performed on the entire test sets.

Table 7: **Generalization to Model Families.** We provide results for the two tasks (Image Classification and VQA) by using the Qwen2.5-VL-3B (Bai et al., 2025).

data is drawn from only a small subset of classes (*e.g.,* 4 out of 200 in ImageNet-R), against *random sampling*, where data is sampled uniformly across classes. Remarkably, even under biased sampling, TTRV yields substantial improvements over the base model.

**Random Rewards.** Shao et al. (2025) recently showed that some models trained with GRPO can exhibit performance gains even when optimized with spurious rewards. As a sanity check, we compare TTRV against such random rewards and report results in Table 5. We find that InternVL (Chen et al., 2024b) models do not appear to benefit from random rewards, suggesting that their improvements under TTRV stem from meaningful reward signals rather than spurious correlations.

**Single Sample TTRV.** To further test whether the gains from TTRV arise from true task enhancement rather than adaptation to the data distribution, we consider an extreme data-scarce setting where adaptation is performed on only a single randomly chosen test example. Results in Table 6 show that even in this case, TTRV yields measurable improvements, lending additional support to our hypothesis.

**Generalization to Model Families.** While our main results in Tables 1 & 2 focus on post-training InternVL (Chen et al., 2024b) models, we also examine whether TTRV extends to other architectures. In Table 7, we present results with Qwen2.5-VL (Bai et al., 2025) and observe consistent gains. These findings suggest that TTRV is not restricted to a single model family and can generalize across diverse VLM architectures.

## 5 LIMITATIONS AND CONCLUSION

**Limitations.** While we empirically show that TTRV enhances task-specific abilities of the base model rather than adapting to the data distribution, we do not yet provide a theoretical explanation for this behavior. Establishing such a foundation remains an important direction for future work.

**Conclusion.** We introduced TTRV, the first test-time reinforcement learning framework for VLMs, where rewards are extracted on-the-fly from unlabeled test data. Specifically, we proposed two complementary rewards: one based on the frequency of model predictions and another that regulates diversity in rollouts. Extensive evaluation on 16 benchmarks spanning object recognition and visual question answering shows consistent improvements over strong base models, even outperforming GPT-4. Beyond empirical gains, our ablations highlight data-efficient properties of TTRV and how it enhances task-specific abilities without explicit supervision, pointing to test-time optimization through RL as a powerful paradigm for bridging pre-training and downstream deployment.

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

APPENDIX

In this appendix, we present additional experiments and explanations that provide further insight and clarity beyond the main manuscript. Section A lists additional implementation details and evaluation protocols. Section B presents a detailed overview of the datasets used in our study. In Section C, we describe the prompts utilized in our experiments. Then in Section D, we present additional ablation studies that highlight further aspects of our method and offer deeper insights into its effectiveness. Finally, Section E includes comprehensive pseudocode to facilitate reproducibility and to help readers gain a clearer understanding of the implementation details.

All experiments were conducted on a machine equipped with 4× NVIDIA A100 and 4× NVIDIA A6000 GPUs. For the review process, we also provide our full codebase (`code.zip`) along with detailed execution instructions in `Readme.md`. The codebase will be released publicly upon acceptance.

## A  ADDITIONAL EXPERIMENTAL SETTINGS

**Implementation Details.**    We apply TTRV independently on each benchmark and report the results in Tables 1 & 2. For optimization, we adopt the AdamW optimizer with a cosine learning rate schedule, setting the peak learning rate to $5 \times 10^{-7}$. During rollout, we generate 32 candidate responses with a temperature of 1.0 for all experiments. The reward hyperparameter $\alpha$ is fixed at 0.75 for all datasets. We cap the maximum prompt length at 7524 tokens and the maximum response length at 1024 tokens. We generally report results using 20 samples in the main table. These samples are randomly sampled from the test data. In the appendix, we also provide a comparison between 20- and 500-sample adaptation, and in the ablation study, we further evaluate the extreme case 1-sample adaptation, where the model adapts to a single example before being evaluated on the full dataset.

**Evaluation Protocol:**    For evaluation, we use greedy decoding (temperature $= 0$) across all datasets, covering both recognition and VQA tasks. We convert the object recognition task to a four-way multiple-choice questioning task, following Gavrikov et al. (2024). For the VQA tasks, we employ the official dataset prompts and append the same multiple-choice instruction to standardize responses. Two exceptions are made: for Capture, we use free-form answers as recommended by Pothiraj et al. (2025), and for MME, we convert yes/no questions into a multiple-choice format. Performance is measured by accuracy against the ground truth for recognition and VQA tasks, while for Capture we report $1-$symmetric mean percentage error (Pothiraj et al., 2025). For all the zero-shot results we do not employ any chain-of-thought prompting (Wei et al., 2022), because that evaluation setting is more fair with the setting employed in our work.

## B  DATASET DESCRIPTION

To comprehensively evaluate our method, we curated a diverse set of recognition and VQA benchmarks that span multiple task-specific challenges. Table 8 provides detailed statistics of the datasets used in our experiments, including both the original test sizes and the number of images retained after preprocessing.

We employed several widely used recognition datasets that test the robustness and generalization capability of models across distribution shifts. Specifically, we included ImageNet (Deng et al., 2009), ImageNet-V2 (Recht et al., 2019), and ImageNet-A (Hendrycks et al., 2021b) to capture generic object recognition in both standard and adversarial settings. In addition, ImageNet-Sketch (Wang et al., 2019) and ImageNet-R (Hendrycks et al., 2021a) were incorporated to examine robustness under edge-based and texture-based distortions, respectively. To further assess fine-grained and material recognition, we used Food101 (Bossard et al., 2014) and DTD (Cimpoi et al., 2014), which emphasize category-level detail and texture variation.

To test higher-level reasoning, we included a wide range of VQA datasets spanning mathematical ability, general understanding, and compositional reasoning. Mathematical reasoning was evaluated using Mathverse (Zhang et al., 2024a) and MathVista (Lu et al., 2024), while Seed (Li et al., 2023a) and MME (Yin et al., 2023) were selected for general multimodal understanding. RealWorldQA (AI, 2024) was used to benchmark models against real-world scenarios, where all images were first standardized to a maximum resolution of $1000 \times 1000$ for consistency across experiments. We also included Capture (Pothiraj et al., 2025) to probe counterfactual reasoning, CRPE (Wang et al., 2024) to evaluate

| Dataset | Used Test Size | Original Test Size | Focus |
|---|---|---|---|
| ImageNet-A (Hendrycks et al., 2021b) | 7,467 | 7,500 | Generic |
| ImageNet-V2 (Recht et al., 2019) | 9,772 | 10,000 | Generic |
| ImageNet (Deng et al., 2009) | 49,032 | 50,000 | Generic |
| ImageNet-Sketch (Wang et al., 2019) | 35,350 | 50,000 | Edges |
| ImageNet-R (Hendrycks et al., 2021a) | 28,506 | 30,000 | Texture |
| DTD (Cimpoi et al., 2014) | 5640 | 5,640 | Edges, Texture |
| Food101 (Bossard et al., 2014) | 25,250 | 25,250 | Fine-grained |
| Resisc45 (Cheng et al., 2017) | 4,500 | 4,500 | Satellite Imagery |
| Mathverse (mcq) (Zhang et al., 2024a) | 1631 | 2180 | Mathematical Ability |
| Mathvista (Lu et al., 2024) | 490 | 1000 | Mathematical Ability |
| Seed (Li et al., 2023a) | 3,881 | 13,991 | General Understanding |
| MME (Yin et al., 2023) | 1576 | 2,370 | General Understanding |
| RealworldQA (AI, 2024) | 765 | 765 | Realworld Understanding |
| Capture (Pothiraj et al., 2025) | 817 | 962 | Counterfactual Understanding |
| CRPE (Wang et al., 2024) | 7575 | 7575 | Compositionality and Halluncination |
| AI2D (Kembhavi et al., 2016) | 2704 | 3090 | Grpah and Chart Understanding |

Table 8: **Statistics of Recognition and VQA datasets** used in TTRV. We drop images with resolution higher than $1000 \times 1000$. Hence, we report both i) the original number of test images and ii) the used number of test images (*i.e.*, those below the $1000 \times 1000$ threshold).

compositionality and hallucination resistance, and AI2D (Kembhavi et al., 2016) to study performance on diagram, graph, and chart-based understanding tasks.

For computational resons, we filtered out images exceeding a resolution of $1000 \times 1000$ pixels across all datasets, retaining only those within this threshold. The reported "used test size" in Table 8 reflects this preprocessing step. In particular, for the RealWorldQA dataset, where image dimensions were highly inconsistent, we explicitly resized all images to $1000 \times 1000$ resolution to ensure compatibility with our evaluation pipeline.

Overall, the curated dataset collection provides a broad coverage of recognition, reasoning, and real-world understanding challenges, allowing us to rigorously evaluate the generalization capability of our proposed approach.

## C  TTRV PROMPT DETAILS

In this section, we provide the prompts used in our experiments. For each dataset, we present a representative example of the prompt employed in our study. While the specific prompts may vary depending on the nature of the question, particularly in VQA tasks, we provide a general outline illustrating the structure and format of the prompts used across different datasets.

- **ImageNet:**

    ```
    <image> \n Look at the given image and identify what it shows.
    Choose the correct answer from the options below and respond
    with only the corresponding option letter (A, B, C, or D).
    Do not include any explanation or extra text.
    \n Options:\nA. neck brace\nB. shopping cart\nC. guillotine
    \nD. garbage truck.
    ```

- **ImageNet-V2:**

    ```
    <image> \n Look at the given image and identify what it shows.
    Choose the correct answer from the options below and respond with
    only the corresponding option letter (A, B, C, or D). Do not
    include any explanation or extra text. \n Options:\nA. cuirass
    \nB. dial telephone, dial phone\nC. beaver\nD. desk.
    ```

- **ImageNet-R:**

    ```
    <image> \n Look at the given image and identify what it shows.
    Choose the correct answer from the options below and respond with
    only the corresponding option letter (A, B, C, or D). Do not
    ```

```
include any explanation or extra text. \n Options:\nA. skunk
\nB. panda\nC. german_shepherd_dog\nD. orangutan.
```

- **ImageNet-S:**

```
<image> \n Look at the given image and identify what it shows.
Choose the correct answer from the options below and respond with
only the corresponding option letter (A, B, C, or D). Do not
include any explanation or extra text. \n Options:\nA. lab coat
\nB. cheetah\nC. ptarmigan\nD. canoe.
```

- **ImageNet-A:**

```
<image> \n Look at the given image and identify what it shows.
Choose the correct answer from the options below and respond with
only the corresponding option letter (A, B, C, or D). Do not
include any explanation or extra text. \n Options:\nA. feather boa
\nB. garter snake\nC. soap dispenser\nD. tank.
```

- **Food101:**

```
<image> \n Look at the given image and identify what it shows.
Choose the correct answer from the options below and respond with
only the corresponding option letter (A, B, C, or D). Do not
include any explanation or extra text. \n Options:\nA. Greek
salad\nB. Red velvet cake\nC. Bibimbap\nD. Pork chop.
```

- **DTD:**

```
<image> \n Look at the given image and identify what texture it
shows. Choose the correct answer from the options below and
respond with only the corresponding option letter (A, B, C, or D).
Do not include any explanation or extra text. \n Options:\nA.
banded\nB. crosshatched\nC. freckled\nD. marbled.
```

- **Resisc45:**

```
<image> \n Look at the given image and identify what it shows.
Choose the correct answer from the options below and respond with
only the corresponding option letter (A, B, C, or D). Do not
include any explanation or extra text. \n Options:\nA. industrial
area\nB. sea ice\nC. circular farmland\nD. golf course.
```

- **Mathverse:**

```
<image> \nPlease directly answer the question and provide the
correct option letter, e.g., A, B, C, D.\nDo not include any
explanation or extra text.\nQuestion: Emile is observing a wind
turbine. The vertical distance between the ground and the tip of
one of the turbine's blades, in meters, is modeled by $H(t)$
where $t$ is the time in seconds. What is the meaning of the
highlighted segment?\nChoices:\nA:The turbine's center is 35
meters above the ground.\nB:The turbine completes a single
cycle in 35 seconds.\nC:The length of the blade is 35 meters.
\nD:The turbine has 35 blades.
```

- **Mathvista:**

```
<image> \nHint: Please answer the question and provide the correct
option letter, e.g., A, B, C, D, at the end.\nDo not include any
explanation or extra text.\nQuestion: In the figure above,
triangle ABC is inscribed in the circle with center O and diameter
AC. If AB = AO, what is the degree measure of angle ABO \nChoices:
\n(A) 15*\\degree\n(B) 30*\\degree\n(C) 45*\\degree\n
(D) 60*\\degree\n(E) 90*\\degree.
```

| | Food101 | DTD | Resisc45 | ImageNet | ImageNetv2 | ImageNetR | ImageNetS | ImageNetA |
|---|---|---|---|---|---|---|---|---|
| InternVL3-2B | 67.19 | 37.24 | 72.28 | 56.00 | 67.43 | 66.01 | 62.19 | 67.92 |
|   TTRV 20 samples | 95.60 | 89.73 | 90.06 | 98.31 | 98.25 | 96.89 | 94.74 | 96.31 |
|   TTRV 500 samples | 96.20 | 89.99 | 93.67 | 98.89 | 98.95 | 97.85 | 95.54 | 96.38 |
| InternVL3-8B | 78.32 | 59.11 | 83.62 | 79.47 | 62.58 | 59.32 | 54.48 | 57.03 |
|   TTRV 20 samples | 87.13 | 77.92 | 89.19 | 91.43 | 85.27 | 63.51 | 53.79 | 81.43 |
|   TTRV 500 samples | 97.20 | 89.37 | 93.82 | 99.31 | 97.24 | 96.88 | 95.03 | 96.86 |

Table 9: **Number of Samples for Adaptation.** Top-1 Accuracy (%) obtained by sampling varying data points from the test data.

- **SEED:**

  <image> \nWhat is the main color of the dress worn by the woman with the density value of [0.6536, 0.5600, 0.7431, 0.7743]? \nChoose the correct answer from the options below and respond with only the corresponding option letter (A, B, C, or D). Do not include any explanation or extra text.\nOptions:\nA. Red\nB. None of the above\nC. Brown\nD. Tan

- **MME:**

  <image> \n Is this photo taken in a place of home office? Please answer with yes or no. Please respond with only the corresponding option letter (A or B). Do not include any explanation or extra text. \n Options:\nA. No \nB. Yes.

- **RealWorldQA:**

  <image> \nHow many lanes are there on the left?\n\nOptions are: \nA. 4\nB. 3\nC. 2\nD. 5\n\nPlease answer directly with only the letter of the correct option and nothing else.",

- **Capture:**

  <image>\nCount the exact number of sunglasses in the image. Assume the pattern of sunglasses continues behind any black box. Provide the total number of sunglasses as if the black box were not there.\nPlease reason step by step, and put your final answer within \\boxed{}.

- **CRPE:**

  <image> \nWhat is the person in front of?\nA. The person is in front of the person.\nB. The person is in front of the tree.\nC. The person is in front of the mirror.\nD. The person is in front of the shelf.\nAnswer with the option's letter from the given choices directly and do not include any explanation or extra text.

- **AI2D:**

  <image> \nWhat is the line that divides the two different plates? \nChoose the correct answer from the options below and respond with only the corresponding option letter (A, B, C, or D). Do not include any explanation or extra text.\n\nOptions:\nA. ground\nB. fault line\nC. dirt\nD. earthquake.

|  | Latency (avg $\pm$ std) | Overhead vs. Normal Inference |
|---|---|---|
| Normal Inference | 25.5$\pm$4.5 s | – |
| *Adaptation:* | | |
| 1 sample | 2.75$\pm$0.43 m | $\approx$+2.7 m |
| 20 samples | 3.77$\pm$0.63 m | $\approx$+3.8 m |
| 500 samples | 1 hr 38 m $\pm$16 m | $\approx$+1 hr 38 m |

Table 10: **Computation Overhead.** Inference and adaptation latency through TTRV. Seconds: s, Minutes: m, Hours: h.

|  | DTD | Imagenet-A | Imagenet-V2 | AI2D | Mathverse |
|---|---|---|---|---|---|
| InternVL2.5-4B | 46.78$\pm$ 0.03 | 90.58 $\pm$ 0.05 | 83.01 $\pm$ 0.03 | 51.73 $\pm$ 0.01 | 52.67 $\pm$ 0.55 |
| w/ TTRV | 81.87$\pm$ 0.80 | 96.09 $\pm$ 0.01 | 96.77 $\pm$ 0.06 | 64.75 $\pm$ 2.34 | 53.59 $\pm$ 0.45 |

Table 11: **Variance of Results.** Results obtained by employing TTRV across 5 independent runs.

|  | Mathvista $\rightarrow$ Mathverse | Food $\rightarrow$ Mathvista | DTD $\rightarrow$ Seed | IN-V2 $\rightarrow$ IN-R | IN-V2 $\rightarrow$ IN-A | IN-A $\rightarrow$ IN-V2 |
|---|---|---|---|---|---|---|
| InternVL2.5-4B | 51.69 | 65.49 | 56.25 | 79.53 | 90.67 | 83.07 |
| w/ TTRV | 52.00 | 67.14 | 59.07 | 95.42 | 96.30 | 96.14 |
| $\Delta$ | +0.31 | +2.52 | +2.02 | +15.89 | +5.63 | +13.07 |

Table 12: **Cross-dataset generalization**. Performance on different dataset combinations, where X $\rightarrow$ Y denotes training on dataset X and testing on dataset Y. "IN" in the table refers to ImageNet.

# D  ADDITIONAL EXPERIMENTS

## D.1  LATENCY VERSUS NUMBER OF SAMPLES

To further substantiate our claims, we conducted experiments aimed at analyzing the adaptability of the model under varying conditions. The first experiment investigates how the model's performance changes when it is allowed to adapt using different numbers of training samples. Specifically, we compare the outcomes when the model is adapted with only 20 samples versus when it is adapted with 500 samples. As shown in Table 9, the results demonstrate a consistent improvement in performance as the number of adaptation samples increases. This suggests that providing the model with a richer set of examples enables it to better align with the target task, thereby yielding higher accuracy and robustness. However, this improvement does not come without trade-offs. Increasing the number of samples also leads to a higher computational burden, both in terms of memory consumption and processing time. This is particularly important in real-world applications where inference latency is a critical factor. To quantify this trade-off, we conducted an additional experiment measuring the latency associated with adaptation on different sample sizes. The results in Table 10, reveal that while larger adaptation sets enhance task performance, they simultaneously increase the time required for inference, thereby highlighting an inherent balance between accuracy and efficiency.

All experiments were conducted using the vLLM inference engine, one of the fastest and most recent frameworks for large language model inference. Despite its efficiency, optimized inference remains an active research area, and ongoing improvements in frameworks such as vLLM are expected to further reduce latency. Moreover, the reported times are dependent on the underlying hardware. Access to more powerful GPUs would likely accelerate both inference and adaptation, thereby reducing the overall time required for these tasks.

## D.2  ROBUSTNESS

To evaluate the robustness of our method, we conducted experiments to measure the variance in its performance. The results are summarized in Table 11. As shown, our method, when evaluated using greedy decoding, demonstrates strong robustness and is only subject to minor variations attributable to hardware and software factors.

### D.3 FURTHER CROSS-DATA GENERALIZATION EXAMPLES

In addition to the results shown in Figure 3, we present challenging cross-dataset evaluation results in Table 12. Our method consistently improves performance across all transfer settings, demonstrating its effectiveness not only in within-domain accuracy but also in transferring knowledge across diverse domains. For instance, training on ImageNet-V2 leads to significant gains when tested on ImageNet-R (+15.89%) and ImageNet-A (+5.63%). Similarly, training on ImageNet-A improves performance on ImageNet-V2 by +13.07%. We also observe positive transfer in mathematical reasoning tasks, with a gain of +0.31% when training on MathVista and evaluating on MathVerse. Notably, even when models are trained on visual recognition datasets and evaluated on VQA benchmarks, such as training on Food and testing on MathVista, we achieve a performance improvement of +2.52%. These results indicate that our approach enhances visual understanding in a way that generalizes well across heterogeneous tasks and domains.

# E PSEUDOCODE

The following pseudocode illustrates the main steps of our method, including rollout generation, reward computation, advantage estimation, and policy update.

**Pseudocode: Test-Time Reinforcement Learning for Vision Language Models**

```python
def inference_time_grpo(model, test_sample, N=32, alpha=0.75, lr=0.001):
    """
    Args:
        model: decoder-based VLM with parameters theta
        test_sample: (x) consisting of image + text prompt
        N: number of rollouts per test sample
        alpha: weight for entropy regularization
        lr: learning rate for policy update

    Returns:
        updated_model: model with adapted parameters
    """

    # 1. Generate rollouts
    responses = [model.sample(test_sample) for _ in range(N)]
    unique_responses = set(responses)

    # 2. Empirical probabilities
    freq = {y: responses.count(y) for y in unique_responses}
    probs = {y: freq[y]/N for y in unique_responses}

    # 3. Compute rewards
    r1 = {y: probs[y] for y in unique_responses} # Frequency-based reward
    H = -sum(p * log(p) for p in probs.values())
    r2 = -H # diversity control reward
    R = {y: r1[y] + alpha * r2 for y in unique_responses} # total reward

    # 4. Convert rewards -> relative advantages
    mean_R = sum(R[y] for y in responses) / len(responses)
    std_R =
        (sum((R[y] - mean_R)**2 for y in responses) / len(responses))**0.5
    if std_R < 1e-8: # avoid divide by zero
        A = {y: 0.0 for y in responses}
    else:
        A = {y: (R[y] - mean_R) / std_R for y in responses}

    # 5. Policy update (GRPO)
    grad_estimate = 0
    for y in responses:
        logprob = model.log_prob(test_sample, y)
        grad_estimate += A[y] * grad(logprob, model.params)

    grad_estimate /= N

    for param in model.params:
        param += lr * grad_estimate[param]

    return model
```

