# OpenReview forum: "TTRV: Test-Time Reinforcement Learning for Vision Language Models"
_ICLR.cc/2026/Conference — ICLR 2026 Conference Withdrawn Submission_

### Official Review · Reviewer_ZgqZ · 2025-10-25

**Soundness:** 3
**Presentation:** 2
**Contribution:** 2
**Rating:** 4
**Confidence:** 4

**Summary:**

TTRV introduces a new paradigm of test-time reinforcement learning (RL) for Vision-Language Models (VLMs). Instead of fine-tuning on labeled datasets, the model self-adapts during inference by performing RL updates based on unsupervised reward signals derived from its own outputs. The authors design two rewards:
1. Frequency reward—encouraging consistent outputs across rollouts, and
2. Diversity control reward—penalizing over-confident or repetitive predictions.
The model uses Group Relative Policy Optimization (GRPO) to perform these updates at inference. This yields significant improvements across 16 datasets (up to +52% accuracy on recognition and +29% on VQA) and sometimes surpasses GPT-4o.

**Strengths:**

1. Novel paradigm:
Shifting RL from training to inference time is a radical and creative idea. It opens up self-improving model behavior at deployment without retraining.
2. Empirical strength: The reported gains—particularly in zero-shot recognition—are large and consistent.
3. General applicability: The method is architecture-independent and compatible with various pretrained VLMs (Qwen2.5-VL, InternVL3, etc.).
4. Interpretability: The frequency and diversity rewards are intuitive and explainable, giving transparency to the adaptation process.
5. Practicality: Test-time adaptation can be done on unlabeled data, an important benefit for real-world deployment.
6. Strong motivation and clear writing:
7. The paper builds its case logically from human analogy—how humans improve from feedback even after “training.”

**Weaknesses:**

1.	Heuristic reward design:
The rewards (frequency/diversity) are empirical and lack theoretical justification. Their combination could yield unstable gradients.
2.	Risk of overfitting:
Per-instance adaptation may bias the model to recent examples, harming generalization.
3.	Computational cost:
Requiring multiple rollouts per sample increases inference time substantially, undermining efficiency.
4.	Limited comparison with test-time training (TTT):
Methods like Tent and Entropy Minimization are not compared, missing key baselines.
5.	No memory mechanism:
The adaptation is local to each test input; there is no evidence of accumulation of learned behavior over time.
6.	Ablation limitations:
The contribution of each reward term is shown but not systematically studied under noise or domain shift

**Questions:**

1.	How does TTRV compare to entropy-based test-time training (e.g., Tent, 2023)?
2.	Does adaptation persist across samples or reset after each inference?
3.	How is reward scaling handled to avoid gradient explosion?
4.	What is the average latency increase per test sample?
5.	Could the method degrade in open-ended generative tasks?

---

### Official Review · Reviewer_oFjH · 2025-10-27

**Soundness:** 4
**Presentation:** 4
**Contribution:** 3
**Rating:** 6
**Confidence:** 4

**Summary:**

This paper presents a framework for test-time RL for VLM models. Unlike traditional RL framework for VLM model that require post-training on a labeled dataset, this paper proposed to directly conduct RL on the testing dataset without any label. To enable this, they proposed to combine a frequency-based and entropy-based reward function. Evaluation performance on a decent amount of image recognition and VQA benchmarks shows the effectiveness of the proposed method.

**Strengths:**

1. Interesting idea to extend TTRL to VLM with entropy loss.

This paper extends TTRL framework to VLMs with an explicit entropy regularization term. Although this might seem to be a small modification, as shown in Table 3, the combination of frequency and entropy reward leads to significantly better performance. I believe this suite of aware would be helpful for future research.

2. Strong performance on the benchmarks.

The proposed method shows consistent strong improvements across a diverse set of VLM benchmarks, indicating this approach is not limited to a single task or model family, as shown in Table 1 and Table 2. This makes the conclusion of the paper more convincing.

3. Extensive experiments with clear ablation studies

This paper provides clear ablations (with/without entropy, different VLM base models, using random rewards) to further prove the effectiveness of TTRV in different perspectives. I think this study would be very interesting to many researchers in the area.

4. Interesting result to show effectiveness with only 1 sample.

Kind of surprisingly, the method also show meaningful improvement even with a single unlabeled sample, suggesting practical value for on-the-fly training for the testing data.

**Weaknesses:**

1. Might be hard to transfer this method to open-ended question answering tasks.

The method's frequency-based reward is best aligned with closed-form answers, where you can meaningfully compute the distribution of answer. For open-ended or multi-valid responses, this reward might have limited usefulness. It would be helpful to discuss possible ways to extend to these tasks and have discussions of this limitation in the paper.

2. Lack of discussion of how to deal with the potential risk of circulating the errors in the model.

Because rewards are derived from the model's own sampled outputs, there is a risk of amplifying initial errors through this self-reinforcement stage and suppressing diversity under strong entropy pressure. It would be helpful to discuss some failure cases of when this happens, or explain why this would rarely happen in real applications.

**Questions:**

How does doing the RL with the same loss on the training data work? Would the model also benefit from this? This would be important for the case of single-example setting of TTRV, because if the method work better with more examples from the training data, there is no point of training it on a single testing example.

---

### Official Review · Reviewer_uzF6 · 2025-10-30

**Soundness:** 2
**Presentation:** 2
**Contribution:** 1
**Rating:** 2
**Confidence:** 5

**Summary:**

The submission proposes TTRV, a test‑time reinforcement learning framework for VLMs. For each test prompt, the model samples candidate responses, forms an empirical distribution over the unique outputs, and uses two unsupervised rewards:

- a frequency reward proportional to empirical response frequency;
- a diversity‑control term that rewards low entropy of the empirical distribution.

The final reward is optimized with GRPO during inference. Experiments on 16 datasets (8 recognition, 8 VQA) report very large gains, e.g., InternVL3‑2B on ImageNet from 56.0% to 98.31% after adapting on 20 unlabeled test samples per dataset, with additional ablations and latency measurements.

**Strengths:**

- Simple recipe with fixed hyper‑parameters ((N=32), (alpha=0.75)), pseudocode, and broad experimental surfac.
- Negative‑control check on random rewards and some cross‑dataset transfer.

**Weaknesses:**

1) Incremental technical contribution. The method is essentially a direct and superficial combination of test-time RL [1] and VLM architectures. The reward formulation mirrors TTRL’s majority-vote reward with a trivial entropy reward. This is far from a substantial conceptual advance.

2) Lack of baseline. There is no self‑consistency/best‑of‑\(N\) baseline (same decoding budget/temperature, etc.). The paper also does not position TTRV against R1‑style VLM RL (MM-EUREKA [2] , Vision-R1 [3], Perception‑R1 [4]). Those works leverage verifiable rewards and report nuanced phenomena (e.g., reward hacking, data quality effects) that bear directly on claims here.

3) Converting recognition to 4‑way MCQ and standardizing VQA to MCQ (Appendix A) changes the task. Massive leaps (ImageNet to ~99%) likely reflect selection‑from‑options under heavy sampling rather than genuine classification ability.

4) Main results focus on InternVL; a small Qwen2.5‑VL check appears only in ablations. Given the 2025 Spurious Rewards [5] findings, strong claims about test‑time RL should be demonstrated on diverse model families in the main table.

[1] TTRL: Test-Time Reinforcement Learning (arXiv:2504.16084)

[2] MM-Eureka: Exploring the Frontiers of Multimodal Reasoning with Rule-based Reinforcement Learning (arXiv:2503.07365)

[3] Vision-R1: Incentivizing Reasoning Capability in Multimodal Large Language Models (arXiv:2503.06749)

[4] Perception-R1: Pioneering Perception Policy with Reinforcement Learning (arXiv:2504.07954)

[5] Spurious Rewards: Rethinking Training Signals in RLVR (arXiv:2506.10947)

**Questions:**

1. How does TTRV differ fundamentally from TTRL beyond the input modality and entropy term?
2. Can you show compute-matched self-consistency baselines (same (N)), and more R1-style VLM baselines?
3. How does TTRV behave on different model families?

---

### Official Review · Reviewer_tPxN · 2025-11-02

**Soundness:** 2
**Presentation:** 3
**Contribution:** 3
**Rating:** 2
**Confidence:** 4

**Summary:**

The paper proposes TTRV, a test-time RL procedure for decoder-style VLMs. For each test prompt, the model samples N outputs, forms an empirical distribution over responses, and uses a frequency‑based reward plus an entropy (diversity) reward inside GRPO to update the policy on the fly.

The paper shows good performance improvement and also studies ablations, sample efficiency, cross‑dataset transfer, and latency. Implementation details are clear.

**Strengths:**

Good extension of test time training to VLM. Simple, general mechanism (self‑consistency + GRPO) that can be bolted onto existing decoder‑VLMs without labels. The pipeline is easy to get.

Large empirical gains across diverse domains, including OOD classification and several VQA benchmarks. The method often remains effective even with 1 adaptation sample.

**Weaknesses:**

The main weakness is experiments miss other test time training baselines. For example, a simple test‑time entropy minimization baseline.
This would be crucial to understand the exact value of TTT on GRPO.

**Questions:**

The paper mention the trade-off between convergence and diversity using two rewards.
Meanwhile, I am still confused why that is the case?

Specifically "explores diverse reasoning modes initially (as encouraged by the frequency-based reward)". Why that is the case? The frequency read gives higher reward to responses that appear more often, would that do the exact opposite?

---

### Note · Authors · 2025-11-13

I have read and agree with the venue's withdrawal policy on behalf of myself and my co-authors.